# How Does Airway Surface Liquid Composition Vary in Different Pulmonary Diseases, and How Can We Use This Knowledge to Model Microbial Infections?

**DOI:** 10.3390/microorganisms12040732

**Published:** 2024-04-03

**Authors:** Dean Walsh, Jennifer Bevan, Freya Harrison

**Affiliations:** School of Life Sciences, University of Warwick, Coventry CV4 7AL, UKf.harrison@warwick.ac.uk (F.H.)

**Keywords:** airway surface liquid, asthma, biofilm cystic fibrosis, host-mimicking media, respiratory infection, sputum, ventilator-associated pneumonia

## Abstract

Growth environment greatly alters many facets of pathogen physiology, including pathogenesis and antimicrobial tolerance. The importance of host-mimicking environments for attaining an accurate picture of pathogen behaviour is widely recognised. Whilst this recognition has translated into the extensive development of artificial cystic fibrosis (CF) sputum medium, attempts to mimic the growth environment in other respiratory disease states have been completely neglected. The composition of the airway surface liquid (ASL) in different pulmonary diseases is far less well characterised than CF sputum, making it very difficult for researchers to model these infection environments. In this review, we discuss the components of human ASL, how different lung pathologies affect ASL composition, and how different pathogens interact with these components. This will provide researchers interested in mimicking different respiratory environments with the information necessary to design a host-mimicking medium, allowing for better understanding of how to treat pathogens causing infection in these environments.

## 1. Introduction

Airway surface liquid (ASL) is a thin film of fluid lining the epithelium of the trachea, bronchi, and bronchioles [1]. ASL comprises two layers [1,2]: the periciliary layer (PCL) which ranges between 5 and 15 µm in depth [3] and the mucus layer (MCL) which can reach 70 µm in thickness [4]. The PCL is a watery layer that bathes the cilia, and may be the only layer of ASL present in healthy adults [5]. However, once the airways are irritated, submucosal and goblet cells in the tracheal and bronchial epithelium release MUC5B and MUC5AC, which form the bulk of the mucus gel-like layer that forms the MCL [1]. This layer is likely to be thicker in some regions than others [6]. In the bronchioles, goblet and submucosal cells are instead replaced by Club cells, which produce MUC5B [7]. ASL and mucociliary clearance are essential in maintaining a very low microbial load and healthy airway functioning [6]. The low viscosity of the PCL allows for ciliary beating, facilitating the movement of the mucus blanket as it is propelled by the tips of the cilia. Cilia have been shown to not only move the mucus sheet of ASL, but also the periciliary sol [1,2,8]. The mucus entraps inhaled particles and microorganisms. The MCL and immobilised particles are directed via cilia to the mouth, where the resulting phlegm is then swallowed or expectorated [8]. Cough clearance also clears mucus independently of cilia activity [9]. An increased height and volume of the PCL is essential for an efficient cough clearance function; conversely, an excessively viscous mucus gel reduces airway clearance via coughing alone [10].

ASL contains numerous microbial growth factors including sugars, protein sources, electrolytes, and lipids [11]. It can take hours to clear airborne pathogens entrapped in mucus; during this time, pathogens can use these components to proliferate. To combat this, antimicrobial peptides including lysozyme, lactoferrin, and defensins are deployed to inhibit pathogen growth [11,12]. Additionally, peptide, amino acid, and sugar transporters can sequester nutrients present in the airway, inhibiting pathogen growth by limiting nutrient availability [12,13]. Despite these innate airway defences, infection does occur, particularly when pre-existing disease or comorbidities compromise these defences. Changes in ASL composition have been shown to predispose individuals to infection [14]. For example, Buonfiglio and colleagues have shown that the increased iron present in the ASL of smokers promoted the growth and biofilm formation of *Staphylococcus aureus* and *Pseudomonas aeruginosa* [14]. Likewise, excess mucin in the ASL mucus of mechanically ventilated patients can impair neutrophil function and consequently increase bacterial growth [15]. 

Numerous host-mimicking growth media have been developed to mimic the nutrition of different host environments, allowing for the monitoring of how these growth environments alter the phenotypes of pathogens [16]. These include artificial urine [17], artificial saliva [18], and synthetic wound fluid [19] for the modelling of urinary tract infections [20], periodontitis [21], and chronic wound infections [22], respectively. For modelling the pulmonary environment, studies investigating both chronic and acute infection of the cystic fibrosis (CF) lung dominate, with a large array of CF sputum-mimicking media described [16,23,24,25,26,27,28,29,30,31]. However, CF sputum is not representative of the ASL and sputum composition in other chronic pulmonary diseases. The concentrations of various macromolecules, sputum viscoelasticity, biomarker profiles, and sputum hydration differ relative to samples from patients with asthma exacerbations, chronic obstructive pulmonary disease (COPD), bronchitis, and bronchiectasis [32,33,34,35,36]. To our knowledge, host-mimicking media simulating these non-CF pulmonary disease environments are yet to be developed, although Ruhleul and colleagues have recently developed healthy lung and healthy sinus-mimicking media [26]. This makes it difficult to mimic the environment of other respiratory infections, such as ventilator-associated pneumonia (VAP). Neglecting the modelling of these infections is concerning.

In VAP, biofilms form around the cuff of the intubated endotracheal tube (ETT) and pathogens use ASL and saliva for growth [37,38]. VAP is defined as a pneumonia occurring after >48 h of mechanical ventilation; incidence varies between countries and hospitals but it is a very common infection in intensive care units with a high mortality rate (25–45%, [39,40]). The global economic impact of biofilms in ventilated patients is an estimated USD920 million per year [41]. VAP has been of particular concern following the coronavirus disease 2019 (COVID-19) pandemic, because the incidence of VAP in COVID-19 patients is much higher (50–80%) [42] than in non-COVID-19 patients (5–40%) [43,44]. 

Due to the lack of in vitro options for simulating the VAP environment, researchers have turned to murine VAP models [45] or ventilated pig models [46] to answer questions regarding the development of VAP. These experiments can be undesirable due to ethical considerations and the notable differences between animal and human ASL [47,48,49,50,51]. PCL composition has been broadly defined (96% water, 1% salts, 1% proteins, 1% lipids, 1% mucus), and the MCL is composed of a mixture of debris and polypeptides tethered by mucins [2,52]. However, the lack of specific information on components and concentrations hampers the development of in vitro VAP models and artificial ASL growth medium. Here, we compile and summarise the findings of studies looking at the composition of both healthy human ASL and the ASL of patients with various lung diseases, aiding researchers in the development of artificial ASL media that can mimic the chemical profile of different pulmonary diseases. This will allow for the development of infection models simulating different pulmonary diseases, providing a better understanding of how to prevent and treat infections in patients with specific pulmonary comorbidities. 

## 2. Mucin

Mucins are large, highly glycosylated (≥50% carbohydrate wt/wt) proteins that form the main macromolecular components of mucus, therefore playing an essential role in the innate defence of the airways [53]. Mucus is a protective lining of healthy airways, whilst sputum is defined as a mucus produced in inflamed airways that also includes cells, inflammatory mediators, bacteria, and DNA derived from inflammatory cell necrosis [54,55]. Within sputum, the secreted mucins MUC5AC and MUC5B dominate [56]. In tracheobronchial surface epithelium, MUC5AC is secreted by goblet cells, whereas MUC5B is produced by submucosal cells [57]. Mucin glycosylation is essential to the rheological and viscoelastic properties of mucus, with mucins in the viscous mucus secretions of people with CF having more sialylated and sulphated O-glycans relative to the mucins in secretions of non-diseased individuals [58]. Changes in mucus viscosity are seen in other chronic bronchial diseases, including COPD and asthma [59]. 

The sputum of patients experiencing CF exacerbations contained ~90% more MUC5AC and ~30% more MUC5B than CF stable patients. Mucus taken from the endotracheal tubes (ETT) of patients with no lung disease actually produced proportions of mucin similar to those found in the sputum of patients experiencing CF exacerbations (12% more MUC5AC in CF exacerbation vs. ETT, 4% less MUC5B in CF exacerbation vs. ETT) [55]. As in previous studies [55,60], Henderson and colleagues also used immunological techniques to show that the sputum of CF patients contained significantly lower concentrations of MUC5B than non-CF sputum. Further investigation showed that human neutrophil elastase, a potent protease found in the inflamed airways of patients with CF [61], VAP [62], COPD, severe asthma, and bronchiectasis [63], could cleave the exposed protein region of MUC5AC and MUC5B, accounting for the lower mucin concentrations in CF sputum when immunological techniques were used [61]. 

The subglottic mucus of mechanically ventilated ICU patients had significantly higher mucus viscosity and concentrations of MUC5B relative to controls. There were no notable differences in MUC5AC concentrations between the two groups. Mucin concentrations showed a significant, weakly positive correlation with the duration of ventilation and patient age [15]. The dominance of MUC5B is also seen in COPD patients, whilst MUC5AC is more prevalent in the airways of smokers [64]. Mucin hypersecretion aids innate airway defences in engulfing inhaled particles, but prolonged mucin hypersecretion, as in critically ill ventilated patients, leads to a viscous accumulation of mucus and the failure of mucociliary clearance. This is due to a combination of factors including the compression of the PCL by a more concentrated mucus layer, inhibiting the lubrication of mucus and causing the adherence of mucus to the epithelium [61]. This is further compounded by the failure of the mucociliary escalator resulting in the redirection of secretions [65]. The stagnation and accumulation of mucus was shown to induce mucin-mediated neutrophil dysfunction and promote bacterial growth [15], particularly as pathogens such as *P. aeruginosa* [66] and *Streptococcus pneumoniae* [67] exploit mucin as a nutrient source.

The pathogen *P. aeruginosa* employs dedicated lectins and adhesins to exploit the altered glycosylation of mucins, whose deviated glycans are used as receptors to enable bacterial adhesion [68]. MUC5AC attenuates *P. aeruginosa* virulence by dispersing biofilms, downregulating type III and VI secretion, suppressing quorum sensing [69] and phenazine biosynthesis [70]. However, it has been shown that in certain lung pathologies such as CF, the antimicrobial properties of mucins are greatly diminished. This has been attributed to altered mucin production, mucin degradation, the altered diffusion of microbial products due to mucus layer dehydration, and changes in mucin glycosylation, therefore altering the signalling potential of mucin [55,60,71,72,73,74]. These alterations to mucin structure and function may explain why *P. aeruginosa* can exhibit abundant quorum-sensing signalling and biofilm formation in CF sputum [75], despite unaltered mucins suppressing quorum sensing and dispersing *P. aeruginosa* biofilms [69]. Pyocyanin, a toxin produced by *P. aeruginosa*, is known to stimulate the hypersecretion of mucin from goblet cells [76]. *S. aureus* utilises serine protease A (SplA) to degrade mucin, therefore facilitating lung invasion in vivo [77]. In the opportunistic fungal pathogen *Candida albicans*, mucin suppresses filamentation, preventing killing by *P. aeruginosa*, which attacks hyphal *C. albicans* [78]. Both *Haemophilus influenzae* and *Streptococcus pneumoniae* can bind to mucin, facilitating the penetration of the mucosal layer, the colonisation of host epithelium cells, and biofilm formation in the airways [79,80]. The negatively charged capsular polysaccharide of *S. pneumoniae* repels the sialic acid-rich mucopolysaccharides of mucin, preventing entrapment [79]. 

Existing CF sputum-mimicking media have included mucin at 5 mg/mL [26,27,31,81], 10 mg/mL [24], or 20 mg/mL [30]. For media attempting to model ventilated airways, it may be appropriate to exceed these mucin concentrations, as Powell et al. found that the average total mucin content of ETT aspirates was ~25 mg/mL [15]. Ruhleul and colleagues also produced a healthy lung medium alongside their CF lung medium. The healthy medium contained 1.2 mg/mL mucin [26], based on the findings of Henderson et al. [61]. Porcine gastric mucin (PGM) was used in most studies [24,26,27,31], although one study did use bovine submaxillary mucin (BSM) [81]. PGM predominantly consists of MUC5AC, although MUC6 and MUC5B are also present [82,83], whereas BSM consists of MUC5B and MUC19 [84]. Due to this, BSM may be more appropriate when trying to model conditions in which excess MUC5B is produced, such as VAP and COPD [15,56]. 

One disadvantage of commercially available mucins is that they can no longer form hydrogels, limiting their rheological and antimicrobial relevance compared to native mucin [85]. Furthermore, commercially available PGM contains various contaminants, including lipids, peptides, amino acids, and metals. This can complicate experiments: a previous study investigating the mucin-enhanced virulence of *Acinetobacter baumannii* in the mouse lung found that mucin-enhanced virulence was at least partially due to iron present in the supplemented PGM [86]. Another study found that iron derived from PGM supplementation raised iron to a more clinically relevant concentration relative to the CF lung [87]. Although BSM is purified in native conditions [88] whilst PGM is not [89], the higher expense and reduced quantity available to purchase of this mucin still makes PGM the most practical mucin source for studies that require high volumes of mucin-containing medium. Despite these limitations, the use of commercially available mucins is likely to continue, as purifying native mucins is labour intensive and produces a low yield [90].

## 3. DNA

DNA present in airway surface liquid and sputum is typically derived from inflammatory responses. Neutrophils are essential for the protection of the airways and produce neutrophil extracellular traps (NETs) in response to invading pathogens. These NETs are composed of DNA complexed with a wide array of antimicrobial proteins [91,92]. Although a key component of innate airway defence, NETs can cause severe lung injury [92,93,94,95]. This is further compounded during mechanical ventilation, which, in addition to being injurious to the lungs in itself, increases NET production [95,96]. The combination of NETs with nucleic acids from dead neutrophils, bacteria, and to a lesser degree, airway epithelia, can lead to the accumulation of a tenacious mucus that obstructs the airways [97,98]. DNA can associate with mucins in the sputum of CF patients [99]; the presence of DNA in mucus makes it much more viscous, reducing mucociliary clearance [60]. Likewise, extracellular DNA has been shown to regulate the expression of numerous virulence and metabolic genes in *P. aeruginosa* and is an essential structural polymer in the biofilm matrix [100]. Matrix DNA is known to aid in the survival of bacteria within biofilms by promoting antibiotic tolerance through the acidification of the environment and shielding *P. aeruginosa* from aminoglycosides [101]. *S. aureus* isolates chronically infecting individuals with CF express higher levels of nuclease activity, allowing *S. aureus* to persist in the neutrophil-rich environment by degrading NETs [102]. 

When developing media to mimic healthy sinuses and healthy lungs, Ruhleul et al. incorporated 0.96 mg/mL of DNA into the final recipe, based on quantities found in the sputum of healthy patients by Henke and colleagues [55]. Numerous formulations of CF-mimicking media contained 4 mg/mL of DNA [26,28,29,31]. Some recipes decreased the DNA concentration to 1.4 mg/mL [24,25] or even 0.6 mg/mL (refs) to better reflect the results shown by Brandt et al. in which the DNA was quantified from a large number of CF sputum samples (ranging from 0 to 9.5 mg/mL) [103]. Deciding on an appropriate DNA concentration for future ASL-mimicking media is difficult, due to the large variance across multiple studies, even those using similar methodologies (Table 1) [15,55,92,103,104]. 

## 4. pH and Ion Concentrations

Airway pH regulates ion transporters, influencing the movement of water and salt between cells and the ASL. Consequently, the effect of pH on the equilibrium between the secretion of chloride ions (Cl^−^) and the absorption of sodium ions (Na^+^) dictates ASL volume and hydration [2,122]. The pH of ASL in healthy individuals is recorded as ranging from 6.78 to 7.1 [106,107]. People with CF, who have mutations in their cystic fibrosis transmembrane conductance regulator (*CFTR*) gene encoding an anion channel, have a reduced ASL pH [122,123], and their airway mucus becomes much thicker and more viscous. Many studies have measured the pH of CF respiratory mucus and collectively found values ranging between 6.0 and 7.4 [16,52,106,107,108,109,124]. Likewise, other disease states feature ASL acidification including pneumonia [111], COPD [112], and chronic lung disease [110]. Reducing pH below 7.0 reduces the activity of ASL antimicrobials, reduces the frequency of ciliary beating, and increases mucus viscosity [123,125]. ASL acidification also impairs *S. aureus.* clearance from human primary bronchial epithelial cells obtained from lobectomies [126] and *P. aeruginosa* clearance from a murine lung infection model [127]. Fungal pathogenesis is highly dependent upon pH, with many fungi, including *Candida* species and *Aspergillus* species able to utilise acidification and alkalinisation mechanisms to facilitate colonisation. However, this is still largely unexplored in the lung environment [128]. Ion concentrations have also been quantified, with elevated concentrations of both Na^+^ and Cl^−^ in CF tracheal aspirates [63,107,109,129]. 

Due to the variance in the reported pH values of CF sputum [16,52,106,107,108,109,124], CF sputum-mimicking media have used various pH values including 6.9 [26,28,29], 7 [31], 6.5 [24,25,30], and 6.8 [16,26,81]. NaCl concentrations in most of these media have remained largely unchanged since they were first incorporated into the artificial sputum medium (ASM) recipe of Ghani and Soothill [24,25,28,29,30,31], though later CF-mimicking media, such as the synthetic CF mucus media (SCFM) developed in the Whiteley laboratory, were used ion concentrations determined by those detected in sputum by Palmer and colleagues [16,81]. Likewise, Ruhleul and colleagues [26] used the works of Goldman et al. to determine the NaCl concentrations in their healthy sinus, healthy lung, CF sinus, and CF lung media [130].

## 5. Sugars and Other Carbon Sources

Glucose is present in the ASL of healthy individuals at ~0.4 mM, approximately 12X lower than its blood concentration [131]. Numerous factors can raise ASL glucose concentrations, including viral infection and diabetes mellitus [114]. Increases in breath glucose are seen in hyperglycaemic diabetes patients and CF patients, with hyperglycaemic CF patients having an even higher airway glucose concentration [113]. High blood glucose has been found to be the single most important factor determining whether patients develop exacerbations of COPD [132], severe COVID-19, or acute respiratory distress syndrome (ARDS); whether ICU patients require ventilation; or whether COVID-19 infection proves fatal [133,134,135,136]. Higher blood glucose concentrations have been shown to result in higher endotracheal glucose concentrations [114]. One study looking at 58 critically ill ventilated patients found that the glucose concentration of bronchial aspirates ranged between 2.7 and 4.4 mM [115]. 

The low glucose concentration in healthy ASLs ensures invading airborne pathogens are deprived of available carbon. This is maintained by tight junctions between airway epithelial cells, limiting the paracellular movement of glucose [137]. Furthermore, human airway epithelia are able to generate a transepithelial glucose concentration gradient, resulting in an ASL with a lower glucose concentration than blood [13]. Using a human airway epithelia culture model, Pezzulo and colleagues were able to show that concentrations of glucose similar to that of healthy ASL limited the growth of *P. aeruginosa* to the point that human airway epithelia were able to kill off the invading inoculum. However, when a hyperglycaemic murine lung infection model was used, increased susceptibility to *P. aeruginosa* was observed, caused by increased ASL glucose concentrations [13]. Likewise, when airway epithelial cells were exposed to pro-inflammatory mediators, ASL glucose concentrations increased due to the enhanced permeability of tight junctions [131]. This is further illustrated by the finding that treating hyperglycaemic mice with metformin, a drug that reduces tight junction permeability, reduced both airway glucose and airway bacterial load [138]. The increased availability of glucose in the ASL of inflamed and/or hyperglycaemic airways could explain the poorer outcomes in diabetic patients with community-acquired pneumonia [139]. Ventilated patients with higher ASL glucose concentrations were significantly more likely to be infected with pathogenic bacteria, particularly methicillin-resistant *S. aureus* (MRSA) [115].

In previously discussed CF sputum-mimicking medium, glucose has been used at 3 mM [16,81] following the quantification of glucose in CF sputum samples [16]. Ruhleul and colleagues used 1.2 mM glucose for both their CF sinus and lung media [26], following the finding that the nasal glucose concentration of CF patients averaged 1.2 ± 0.9 mM [140]. When designing media to mimic healthy airways, glucose concentrations of ~0.4 mM would be most appropriate, as this has been shown to be the resting glucose concentration of ASL in healthy individuals [131]. When designing media modelling of infected, inflamed, or ventilated airways, using 2–4.4 mM glucose may be more appropriate based on the findings of previous studies [113,115].

Little research has investigated other carbon sources in ASLs. One study found that 15 mM fructose induced a higher growth of *S. aureus* on human epithelial cells relative to 20 mM glucose; however, no studies have yet quantified fructose in human ASL [141]. Mucin degradation can provide an additional source of sugars and amino acids to act as carbon sources. Mucins are comprised of numerous monosaccharides, including N-acetylglucosamine (GlcNAc; 32% of mucin dry weight), galactose (29% dry weight), sialic acid, fucose, and N-acetylgalactose (GalNAc) [58]. The increased glycosylation and sulfation of mucins has been observed in both CF and VAP [58,142], whilst a positive correlation has been observed between airway-infection severity and the increased sialylation of mucins [143]. The increase in mucin modification in diseased states consequently means that higher sugar concentrations are available for bacteria to use as carbon sources following mucin degradation [58,144,145,146]. 

*S. aureus* isolates that can better utilise free sialic acid, such as isolates deficient in the Agr quorum-sensing system, are better adapted to the lung and are associated with chronic lung infection. Furthermore, sialic acid utilisation upregulates the production of the siderophore staphyloferrin, increasing iron acquisition [147]. Whilst *S. aureus* is incapable of liberating sialic acid from mucin itself, sialic acid can be cross-fed to *S. aureus* by residents of the airway microbiota, such as Streptococci or anaerobes [148]. Various respiratory pathogens use free sialic acid to undergo sialylation for the purpose of immune evasion. Sialylated *P. aeruginosa* are able to impede their trafficking to lysosomes when phagocytosed, persisting and replicating intracellularly in macrophages [149]. Furthermore, the binding of sialylated *P. aeruginosa* to neutrophils can suppress neutrophil activity, reducing ROS levels, NET formation, and elastase release [150]. Likewise, nontypeable *Haemophilus influenzae* (NTHi) sialylates lipooligosaccharides, protecting it from IgM and complement-mediated killing [151]. The presence of both sialic acid and GlcNAc has been shown to increase the biofilm formation of *H. influenzae* [152]. The sputum environment also induces GlcNAc catabolism by *P. aeruginosa*, upregulating the production of the phenazine antimicrobial pyocyanin, potentially mediating interspecies competition in the lung environment [146]. The inclusion of these additional carbon sources in host-mimicking media may be important to better represent how respiratory pathogens adapt to the respiratory environment. (Particularly if the medium includes mucin and the organisms of interest are incapable of liberating these carbon sources from mucin). For these reasons, some sputum-mimicking media have incorporated GlcNAc [26,81], galactose, and sialic acid [26].

## 6. Amino Acids 

ASL contains numerous protein sources, including mucins and antimicrobial peptides, which are essential in innate lung defence [12]. To prevent excessive accumulation and subsequent airway obstruction by proteins and mucus, enzymatic degradation within the airways is employed [153]. This provides a wealth of peptides and amino acid sources within the ASL, which are in turn sequestered by various transporters in the lung epithelium. This cycle of protein degradation and peptide transport ensures that there is a constant, if limited, presence of amino acids within healthy ASLs [12]. The acquisition of branched-chain amino acids has been shown to aid in the colonisation of the nasopharynx and lungs by *S. pneumoniae* [154]. Amino acids are abundant in CF sputum, with *P. aeruginosa* preferring the assimilation of amino acids over sugars [155]. This leads to the emergence of *P. aeruginosa* mutants that are auxotrophic for various amino acids [156,157]. The catabolism of histidine has also been shown to been shown to be essential for *A. baumannii* virulence in a murine lung model [158]. Little research has been carried out to identify and quantify specific amino acids present in non-CF ASL. It has been noted that the amino acid of healthy lungs is lower than that of CF patients [116]. Likewise, total amino acid content was higher in CF patients suffering exacerbations than in clinically stable patients [117]. Higher levels of free amino acids are also observed in patients suffering from ARDS [159,160]. The increase in available amino acids in ASL during respiratory disease is hypothesised to be due to reduced amino acid transporter activity [12]. 

Amino acids have been fundamental for the development of effective ASM [16,24,25,26,27,29,30,31,81]. The earliest formulations of artificial sputum medium did not include amino acids [28], and this greatly impaired the biofilm formation of respiratory pathogens [161]. After adapting the findings by Ghani and Soothill [28], Sriramulu et al. began adding selected amino acids to ASM at equal ratios at 250 mg/L, and this was further implemented in subsequent ASM [24,27,29,31]. Conversely, some studies added casamino acids at 5 g/L [25,27] or 7.225× essential amino acids and 14.45× non-essential amino acids [30] instead of adding individual amino acids. Palmer et al. quantified free amino acids in CF sputum supernatants and found relatively consistent ratios of animo acids in samples from different patients, despite differences in overall concentrations. Following this, studies began to incorporate individual amino acids at a ratio reflecting this [16,26,81]. In Ruhleul et al.’s healthy sinus and healthy lung media, amino acid concentrations were determined by the findings of Schwab et al., who determined the amino acid content of non-CF airway secretions [26,162]. Whilst research into the amino acid concentrations in sputum has greatly benefitted the development of CF sputum-mimicking media, and to a lesser degree, media simulating healthy ASL, there is extremely little data on the amino acid concentrations of airways in other disease states, such as the ASL of ventilated patients. It would be logical to assume that the amino acid concentrations in ventilated ASL are higher than healthy ASL, as both people with CF and ARDS have higher sputum amino acid contents than healthy people [159,160,163]. 

## 7. Lipids

Until recently, lipids in the airways were seen only as an energy source or as essential structural components for membranes. Now, lipids are acknowledged to play key roles in mediating the airway immune response through various signalling mechanisms [164,165,166]. The lipid content of healthy ASL has broadly been quantified as approximately 1% of total ASL, and approximately a quarter of all solid content found within healthy ASL [2,52,161]. Likewise, lipids constituted approximately 30% of the dry material of sputum from patients with CF or asthma. Phosphatidlycholine has been identified as the main lipid present in sputum. However, sputum does include other lipid components including cholesterol, triglycerides, ceramides, sphingomyelin, and more [167]. Phosphatidylcholine is thought to act as a biosynthetic intermediate in the formation of phosphocholine-substituted structures on the surface of *H. influenzae* and *S. pneumoniae* [168]. Phosphocholine-modified lipoteichoic acids and teichoic acids of *S. pneumoniae* increase the cell adherence and invasion of the lung [169], whilst phosophocholine-modified lipopolysaccharides are associated with an increased persistence of *H. influenzae* on the airway mucosal surface [168,170]. Phosphatidylcholine utilisation via *P. aeruginosa* in mouse lung infection models increases the pathogen’s fitness, competitiveness, and aids *P. aeruginosa* adaptation to the lung environment [171]. Phosphatidylcholine and cholesterol in pulmonary surfactant have been shown to drive type-3 fimbria-mediated biofilm formation in *Klebsiella. pneumoniae* [172]. Lecithin, provided by egg yolk emulsion, has been the predominant lipid source in artificial CF sputum media [24,25,27,28,29,30,31], and has been universally used at 0.5% (v/v) in these formulations. Dioleoyl phosphatidylcholine has also been used as a lipid source at a concentration of 100 µg/mL [81]. The use of egg yolk lecithin would be more appropriate to provide an array of different lipids; however, this would need to be balanced with the likely diversity in composition between different suppliers and batches [173]. 

## 8. Antimicrobial Peptides and Enzymes

Antimicrobial peptides are released from epithelial cells, submucosal cells, resident and recruited macrophages, and are transported from plasma [174]. The most numerous of these antimicrobials in the airways are lysozyme and lactoferrin [118]. Lysozyme degrades peptidoglycan present in bacterial cell walls, causing lysis [175]. Lactoferrin functions by binding to iron, depriving bacteria of essential nutrition [176]. Lactoferrin can also permeabilise Gram-negative bacteria through interactions with lipopolysaccharide on the cell surface [177]. Concentrations of these antimicrobials can change depending on lung health. For instance, CF has been shown to increase concentrations of lactoferrin in bronchoalveolar lavage fluid (BALF) [61]. One study found no significant difference in lactoferrin concentration between smokers and non-smokers. However, because the BALF of smokers contained higher levels of iron relative to lactoferrin, these conditions were better able to stimulate bacterial growth and biofilm formation [14]. Sagel et al. also detected significant increases in lactoferrin and lysozyme in the BALF of culture-positive CF patients vs. culture-negative CF patients [118].

Other antimicrobial enzymes associated with neutrophils and NETs include myeloperoxidase, which is complexed with the DNA of NETs. Myeloperoxidase catalyses the formation of hypochlorous acid, which in turn kills invading pathogens [178]. Myeloperoxidase concentrations have been shown to increase in ARDS patients, VAP patients, and patients with both conditions (significantly so in the latter two groups) relative to patients with neither condition [92]. Other proteolytic enzymes released from neutrophils include human neutrophil elastase (HNE) and matrix metalloproteases (MMPs). VAP patients have been reported to have significantly elevated levels of HNE, MMP-8, and MMP-9 compared to non-VAP patients [179]. Another study also found significantly elevated levels of HNE in mechanically ventilated ICU patients compared to newly intubated controls [15].

## 9. Metals

Numerous metals are of significant importance in the airways. Calcium (Ca), magnesium (Mg), manganese (Mn), copper (Cu), and zinc (Zn) are important for inflammation [180]. Both Cu and Zn are essential for the functioning of anti-inflammatory superoxide dismutases [181]. Whilst these biometals act as essential co-factors for numerous host enzymes, they can also impact disease severity [182]. Iron (Fe) can cause significant lung damage by generating reactive oxygen species and oxidative stress, and increase bacterial virulence [183]. An iron-enriched lung environment impairs neutrophil function and inhibits the antimicrobial effects of lactoferrin and transferrin through binding saturation, all whilst providing pathogens like *P. aeruginosa* with the nutrition to readily replicate [182,184]. Likewise, excess Mg has been shown to induce neutrophil dysfunction, preventing both phagocytosis and oxidative bursts [185]. These metal ions are suspected to originate from vascular leakage, defects in ion channels, release from necrotic host cells or the lysis of invading microorganisms [183].

Increased metal concentrations have also been detected in numerous inflammatory pulmonary diseases (Table 2) [14,183,186]. Gray et al. quantified Zn, Fe, Cu, and Mn in the sputum of healthy individuals and patients with asthma, COPD, CF, and non-CF bronchiectasis [186]. Concentrations of Cu and Fe were higher in all disease groups compared to healthy controls, whist Zn was higher in all groups except asthmatic patients, and Mn was higher in all but COPD patients. Sputum Zn concentrations also had a strong positive correlation with the presence of lung inflammatory markers including calprotectin, IL-8, and myeloperoxidase [186]. Smith and colleagues found increases in Ca, Mg, Zn, Fe, and Cu concentrations in both CF and bronchiectasis patients relative to healthy controls. They also found that Zn, Fe, and Mg concentrations positively correlated with IL-8 [183], agreeing with the findings of Gray et al. [186]. Fe concentrations were also found to be approximately 4X higher in the BALF of smokers compared to non-smokers [14], whilst smoking status has been found to have little effect on the Zn and Fe concentrations of COPD patients [186]. 

Metals have not been incorporated into many synthetic CF media [24,25,27,28,29,30,31]. SCFM formulations do include FeSO_4_, CaCl_2_, and MgCl_2_ to provide a pool of metal ions [16,81], with Fe concentrations derived from the two studies by Stites et al. [187,188]. Ruhleul and colleagues used the findings of Smith et al. [183] for the basis of the metal concentrations found in their healthy sinus and lung media, as well as in their CF sinus and lung media [26]. Quinn et al. instead incorporated ferritin into their growth medium to better reflect the iron sources present in cystic fibrosis sputum [30].

## 10. Polyamines

Consecutive enzymatic reactions convert L-ornithine into the polyamines putrescine, spermidine, and then spermine [120]. Maintaining normal polyamine concentrations has been associated with preserving numerous essential cellular functions, including cell proliferation and differentiation, ion channel function, and protection against oxidative stress [189]. Polyamines are also synthesised by bacteria for the purposes of altering surface charges to aid antimicrobial and oxidative stress tolerance, protection against the phagolysosome, biofilm formation, and iron and free radical scavenging [172,190]. Polyamines can even be used as a sole carbon source by *P. aeruginosa* [191]. 

Polyamine concentrations in the lung have been shown to vary depending on disease status. For instance, significant increases in putrescine are observed in CF exacerbation patients relative to stable patients [192]. Furthermore, stable CF patients had significantly higher spermine concentrations relative to healthy controls. In addition, patients experiencing pulmonary exacerbation had significantly higher concentrations of both putrescine and spermidine relative to both healthy and CF-stable patients [120]. Metabolomic analyses also found 2–5 log fold increases in polyamines including spermine and spermidine in the endotracheal aspirates of VAP patients compared with samples taken pre-intubation [193]. Significantly higher levels of spermidine have also been observed in smokers and patients with COPD [194]. The elevated levels of polyamines in diseased airways can be exploited to *P. aeruginosa*, which can scavenge free polyamines from the environment [195] and utilise them to facilitate antimicrobial tolerance [195,196]. Polyamines have not yet been widely used in synthetic ASL formulations, though Ruhleul et al. have incorporated putrescine, spermidine, and spermine into formulations of healthy sinus, healthy lung, CF sinus, and CF lung media [26].

## 11. Serum Albumin

Albumin enters sputum through the vascular leakage that accompanies inflammatory pulmonary disease [119]. Albumin in ASL contributes to mucus plugging due to albumin acting as an alternative substrate for neutrophil proteases, and consequently inhibiting mucin degradation. This was evidenced by significantly higher proportions of albumin-degradation products found in the mucus of patients experiencing exacerbations of asthma. Furthermore, albumin can increase the viscoelasticity of airway mucus, impeding mucus clearance [197]. Some bacteria, including *S. pneumoniae* [198] can bind to albumin, exploiting albumin-derived fatty acids as a source of nutrition [199]. Yeast form *C. albicans* can also aggregate around albumin. This facilitates biofilm formation, decreases the penetration of antifungals through *C. albicans*-albumin plaques, and increases protection against phagocytic attack [200]. Albumin also neutralises the cytolytic toxin candidalysin produced by *C. albicans*, potentially reducing the cell damage induced by this pathogen [201]. In *P. aeruginosa*, albumin induces the expression of iron-controlled genes [202] and quenches the homoserine lactone quorum-sensing signal, attenuating the killing of *S. aureus.* via *P. aeruginosa* exoproducts [203]. Albumin levels are significantly elevated in both exacerbating and stable CF patients compared to patients with COPD or healthy individuals [119]. Another study also showed that sputum-producing CF patients produced nearly double the albumin found in the sputum of healthy controls [121]. Bovine serum albumin (BSA) has been incorporated into some previous sputum-mimicking media at 10 mg/mL [24,25]. Ruhleul et al. used BSA concentrations of 0.5 mg/mL for healthy sinus media, 1.5 mg/mL for healthy lung media, and 7 mg/mL for both CF sinus and CF lung media [26]. These concentrations were derived from a study that mimicked the environments for sinus colonisation and pneumonia using *S. pneumoniae* [144].

## 12. Airway Surface Liquid Component Interactions

Understanding how the components of an ASL-mimicking medium interact with each other is essential for interpretating results acquired using these host-mimicking models. PGM has been shown to inhibit the activity of both salivary lysozyme and hen egg-white lysozyme. In total. 2 mg/mL PGM inhibits approximately 40% of the activity of hen egg-white lysozyme, and 20% of the activity of unstimulated whole saliva lysozyme [204]. Mucin concentrations may exceed 2 mg/mL in many media seeking to mimic conditions such as CF exacerbations and VAP, so in these particular conditions, lysozyme inhibition via PGM is likely to be higher. An ASL-mimicking medium would ideally be pH 6–7, which is within the optimal pH range of lysozyme (pH 6–9) [205]. However, pH further impacts the degree of lysozyme inhibition via PGM, with lysozyme activity decreasing by 38.4% at pH 6, and 36.7% at pH 7, in the presence of PGM. Furthermore, many airway-mimicking models would be incubated at 37 °C, but hen egg-white lysozyme activity is inhibited by 45%, and unstimulated whole saliva lysozyme activity is inhibited by 20% after being incubated at 37 °C for 30 min with PGM [204]. Unlike PGM, BSM did not inhibit lysozyme activity [204], perhaps due to its increased purity relative to PGM [87]. Therefore, BSM may be the most appropriate mucin choice for researchers seeking to factor in the effect of lysozyme into their experiments. GlcNAc, which has previously been added to certain airway-mimicking media [26,81], may also lead to a decrease in lysozyme activity, because it is a competitive inhibitor of the enzyme [205,206]. Lysozyme is also known to bind to DNA, forming highly charged lysozyme-DNA complexes that alter nucleic function and make lysozyme a potent antiviral [207,208]. This antiviral activity is distinct from the catalytic mechanism responsible for the antibacterial activity of lysozyme [208]. It is currently unclear whether the presence of high levels of extracellular DNA in airway-mimicking media will alter lysozyme activity. 

Albumin is able to bind and transport many endogenous and exogenous compounds. Due to this, serum albumin is also capable of binding to many other components that would be found in an ASL-mimicking medium, including fatty acids [209], DNA [210], polyamines [211], and metals [212]. Electrostatic interactions between PGM and BSA have been shown to produce mucin–albumin complexes. The formation of these complexes is highly pH-dependent, with mucin and BSA unable to bind at pH 3, but able to bind at pH 7.4, close to the pH used in airway-mimicking environments; BSA and mucin not only bind but cause changes to the secondary structure of mucin. The formation of mucin–BSA complexes alters the microrheology of the mucus layer in individuals experiencing cystic fibrosis, asthma, and ARDS, resulting in a mucus with much lower diffusivity [213]. The formation of mucin–BSA complexes is also facilitated by Ca^2+^ ions [213], which are provided by CaCl_2_ in numerous airway-mimicking media [16,26,81]. PGM has been found to bind lipids including free fatty acids, cholesterol, sphingomyelin, and phospholipids; these bound lipids shield mucin from attack by oxygen radicals [214]. Mucins, including commercially available PGM, have been shown to bind to metals, including iron [215,216] and zinc [215], with iron competitively inhibiting the binding of other metals to mucin [215]. Therefore, any PGM used in ASL-mimicking media may act as a sink for metal and lipid sources added to the medium. Any polyamines added to ASL-mimicking media could also sequester metal ions present in the medium [217]. Furthermore, the polycationic nature of polyamines can result in polyamine aggregation to extracellular DNA present in the growth medium [218], potentially limiting the uptake of polyamines by pathogens grown in ASL-mimicking environments.

## 13. Lung Microbiome

Much like the composition of ASL, the lung microbiome can dynamically change in response to disease, and is therefore an important factor to consider when designing models to mimic specific pulmonary diseases. The lung microbiome is not the core focus of this review, but more in-depth insights into the lung microbiome have been summarised comprehensively in other reviews [219,220]. *Streptococcus*, *Veillonella*, and *Prevotella* dominate the oral microbiome and are also among the most common genera in the lung microbiome of healthy individuals [220,221]. In the airways of individuals with asthma, COPD, and CF, these commensal species have a reduced abundance, and this is accompanied by an increase in pathogenic taxa including *Pseudomonas*, *Staphylococcus*, and *Haemophilus* [222,223,224,225,226,227]. Patients with VAP display a higher bacterial load, with reduced diversity in tracheal aspirates compared to controls. In VAP patients, *Pseudomonas.* and *Corynebacterium* were more abundant, whilst the presence of *Prevotella* and *Streptococcus* has been associated with patients that do not go on to develop VAP [228]. Numerous pathogens reside within the microbiome of protracted bacterial bronchitis patients, including *H. influenzae*, *S. aureus*, and *S. pneumoniae* Enrichments of commensal species of the genera *Prevotella*, *Neisseria*, and *Streptococcus* are also observed [229]. The impact of metabolites from the lung microbiome on lung disease is still largely unexplored, and will in time undoubtedly reveal important metabolites that should be included in ASL-mimicking media. However, the limited study so far into this area does reveal how microbiome and metabolomic data can be used for the development of models simulating different outcomes of a given disease. For instance, in the case of COPD, *Streptococcus*, *Neisseria*, and *Veillonella* in conjunction with metabolites such as polyamines, glycerophospholipids, and glycosphingolipids are associated with COPD and poorer lung function. Conversely, the presence of *Prevotella* with tyrosine and sialic acid are associated with either fewer COPD symptoms or no COPD [230]. As research linking the lung microbiome and lung metabolome with pulmonary disease progresses, the development of host-mimicking models will improve. Media should be assessed for their ability to support a microbiome typical of the condition they seek to model. This will facilitate not only the improved modelling of different pulmonary diseases, but will also allow for the simulation of how pathogens react in nutritional environments associated with different disease outcomes. 

## 14. Future Developments and Conclusions

The composition of human ASL is complex and changes depending on lung health (summarised in Figure 1). Significant research into the CF lung has informed the development of CF sputum-mimicking media. Pathogens grown in these media display gene-expression profiles [16,26,81], the induction of mucoid phenotypes, and biofilm growth resembling that of pathogens growth in CF sputum [24,28,31]. In contrast, there has been only a single attempt to model the ASL of healthy individuals [26], and the development of media mimicking other inflammatory pulmonary conditions is completely lacking. The main reasons for this are, first, the relative lack of research investigating ASL composition in conditions other than CF, and, second, the fact that existing research often focuses on just one particular ASL component, such as amino acids or mucin [142,159,160]. This is partly explained by the relative ease of obtaining large volumes of sputum or BALF from people with CF, who historically have expectorated large quantities of sputum and undergone BAL as part of their medical treatment. Collecting samples from the thin layer of ASL present in healthy people is much harder; although the studies we have cited in this review used a range of methods to attempt this, including a variety of immunologic, chromatographic, and spectroscopic methods. Here, we seek to provide researchers with a comprehensive review of research investigating ASL and sputum composition across a spectrum of inflammatory pulmonary diseases, including ARDS, COPD, and VAP. Whilst CF and CF-mimicking media still feature heavily throughout, we hope this review will act as a reference point for researchers to see how information obtained through numerous studies can transition to the development of lung-mimicking media. 

## Figures and Tables

**Figure 1 microorganisms-12-00732-f001:**
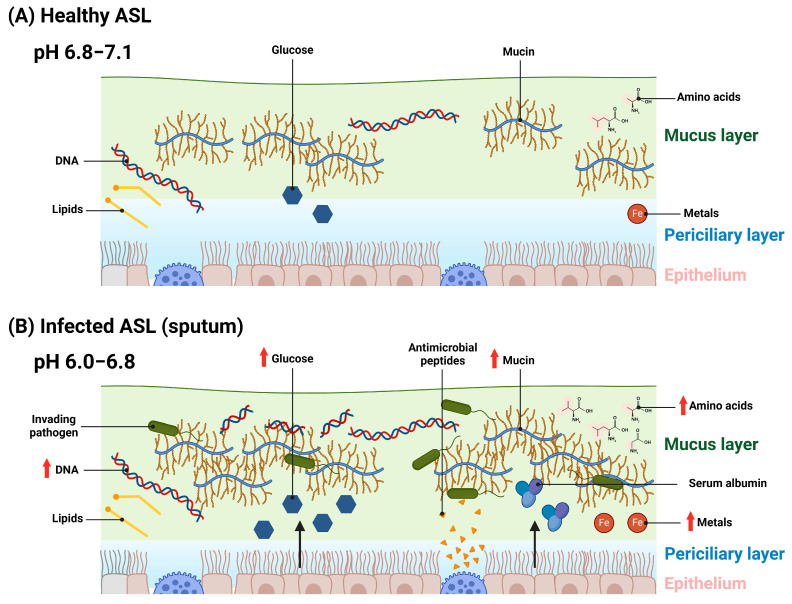
Schematic of the constituent components of healthy and infected ASL (sputum). (**A**) Healthy ASL, composed of a periciliary layer which bathes ciliated epithelium. Above the periciliary layer is a mucus layer, predominantly composed of mucin. Other components in the ASL include lipids, low levels of glucose, amino acids, metals, and DNA. (**B**) ASL infected with invading pathogens. In response to this, more mucin is produced to entrap these pathogens. Antimicrobial peptides are also produced by epithelial and submucosal cells to kill entrapped bacteria and fungi. DNA is present in inflamed ASL in higher amounts due to DNA released by dead host cells and microorganisms, as well as DNA released by neutrophils during the production of neutrophil extracellular traps. Vascular leakage introduces serum albumin into the ASL and causes an increase in ASL glucose concentrations. Airway inflammation has also been shown to result in higher levels of amino acids and metals being present in ASL. ASL pH becomes acidified in the ASL of inflamed airways, this alters ion concentrations and causes ASL dehydration, resulting in a thinner periciliary layer. Created with BioRender.com.

**Table 1 microorganisms-12-00732-t001:** Concentrations of different ASL components detected in different airways pathologies.

Component	Average Concentrations	Conditions	Detection Method
Mucin	2.7 mg/mL [61], 6.5 mg/mL [61], ~5 mg/mL [15], ~20 mg/mL [15].	Healthy [61], CF exacerbation [61], newly intubated elective laryngoscopy patients [15], ICU patients mechanically ventilated for at least 4 days [15].	Size exclusion chromatography/differential refractive index [61], enzyme-linked immunosorbent assay of subglottic samples [15].
DNA	0.96 mg/mL [55], 6.7mg/mL [55], 5.2 mg/mL [55], 20 ng/mL [15,92], 40 ng/mL [15], ~100 ng/mL [92], ~250 ng/mL [92], 0.7 µg/mL [104], 3.2 µg/mL [104], 5.4 µg/mL [104], 2 µg/mL [105], 10 µg/mL [105], 416% more DNA by area in CF sputum compared to asthma and chronic bronchitis sputum [60].	Healthy [55,92,105], stable CF [55,105], CF exacerbation [55], non-ICU patients [15], ICU patients [15], VAP [92], ARDS [92], VAP and ARDS [92], non-CF patients [104], infants with CF [104], older CF patients [104].	Microfluorimetry [55], fluorometric assays [15], colorimetric assays [92], Hoechst dye-binding assay [104], Quant-iT PicoGreen assay [105], confocal microscopy [60].
pH	6.2–7 (nasal) [106], 7.1 (lower airways) [106], 6.78 [107], 7.18 [108], 6.57 [108], 6.97 [109], 6.58 [110], 6.62 [110], 6.72 [111], 6.61 [110], 6.89 [112].	Non-CF [106,107,108,109,110], CF [106,108], pneumonia [110,111], chronic lung disease [110], acute exacerbation of COPD [112].	Monocrystalline antimony catheter [106], in-gold combined pH-glass electrode [106], fluorescent indicators on freshly excised human bronchi [107,109], fluorescent indicators on nasal biopsies [108], pH electrode [110], pH strips [112].
Glucose	0.4 mM [113], 1 mM [114], 4 mM [114], 1.2 mM [113], 2 mM [113], 3.5 mM [115].	Healthy [113], viral infection [114], hyperglycaemic diabetes [114], CF [113], CF and diabetes [113], mechanically ventilated patients [115].	High-performance anion-exchange chromatography with pulsed amperometric detection [113], glucose oxidase sticks [115].
Amino acids	2.52 mg/mL; total [116], 5.7 mg/mL; total [116], 12.3 mM; total [117], 18.2 mM; total [117], 0.42 nmol/mg; alanine [47], 2.2 nmol/mg; asparagine [47], 0.42 nmol/mg; glutamine [47], 1.06 nmol/mg; glycine [47], 0.43 nmol/mg; lysine [47], 0.13 nmol/mg; valine [47].	Healthy [116], CF [116,117], CF exacerbation [117], healthy tissue from lobectomies of lung cancer patients [47].	Thin layer chromatography [116], high-performance liquid chromatography [117], nuclear magnetic resonance [47].
Lysozyme	3.9 µg/mL [118], 9.1 µg/mL [118].	Culture-negative CF patients [118], culture-positive CF patients [118].	Lysozyme activity assay [118].
Lactoferrin	5 µg/mL [61], 9 µg/mL [61], 3.0 µg/mL [118], 22.3 µg/mL [118].	Non-CF [61], CF [61], culture-negative CF patients [118], culture-positive CF patients [118].	Immunologic techniques [61], enzyme-linked immunosorbent assay [118].
Ferritin	0.2 µg/mL [119], 2.4 µg/mL [119], 3.6 µg/mL [119], 0.6 µg/mL [119].	Healthy [119], CF [119], CF exacerbation [119], COPD [119].	Microparticle enzyme immunoassay [119].
Putrescine	11.91 µmol/L [120], 6.18 µmol/L [120], 96.02 µmol/L [120], 20.59 µmol/L [120].	Healthy [120], CF stable [120], CF exacerbation pre-antibiotic treatment [120], CF exacerbation post-antibiotic treatment [120].	High-performance liquid chromatography [120].
Spermine	0.22 µmol/L [120], 1.71 µmol/L [120], 7.32 µmol/L [120], 1.35 µmol/L [120].	Healthy [120], CF stable [120], CF exacerbation pre-antibiotic treatment [120], CF exacerbation post-antibiotic treatment [120].	High-performance liquid chromatography [120].
Spermidine	0.88 µmol/L [120], 1.62 µmol/L [120], 0.78 µmol/L [120], 0.62 µmol/L [120].	Healthy [120], CF stable [120], CF exacerbation pre-antibiotic treatment [120], CF exacerbation post-antibiotic treatment [120].	High-performance liquid chromatography [120].
Serum albumin	0.1 dg/L [119], 0.4 dg/L [119], 0.7 dg/L [119], 0.2 dg/mL [119]. 127.4 µg/mL [121], 244.4 µg/mL [121].	Healthy [119,121], CF [119,121], CF exacerbation [119], COPD [119].	Rate immunophelometry [119], competitive radioimmunoassay [121].

**Table 2 microorganisms-12-00732-t002:** Concentrations of metals (µg/L) in different pulmonary disease across different studies.

Metal	Healthy	Asthma	COPD	CF	Bronchiectasis	Smoker
**Zn**	15.35 (10.4–25.6) ^a^ 179 (103–597) ^b^ 40.45 (20.99) ^c^	12.7 (7.2–41.4) ^a^	25.4 (9.8–50.7) ^a^	135.3 (54.2–209.6) ^a^ 1285 (678–1811) ^b^	111.3 (46.1–150.7) ^a^ 537 (401–838) ^b^	48.16 (35.06) ^c^
**Fe**	13.5 (8.6–21.5) ^a^ 0 (0–37) ^b^ 6.38 (9.12) ^c^	30 (6.9–35.3) ^a^	21.3 (3.1–35.6) ^a^	56.9 (24.3–115.3) ^a^ 797 (398–1292) ^b^	54.2 (22.7–91.6) ^a^ 1075 (862–1324) ^b^	23.37 (28.47) ^c^
**Mn**	0 (0–0.25) ^a^ 5 (2–9) ^b^ 0.12 (0.16) ^c^	0.8 (0.2–1.7) ^a^	0 (0–0.7) ^a^	0.3 (0.1–0.8) ^a^ 6 (4–17) ^b^	0.6 (0.2–1.3) ^a^ 6 (4–10) ^b^	0.21 (0.24) ^c^
**Cu**	8.6 (3–16.4) ^a^ 106 (55.3–196) ^b^ 4.77 (4.9) ^c^	15.2 (8.6–29.5) ^a^	15.2 (12.2–22) ^a^	19.5 (14.5–30.1) ^a^ 173 (128–257) ^b^	15.7 (10.9–33.3) ^a^ 226 (130–314) ^b^	4 (2.26) ^c^
**Ca**	45,000 (28,000–58,000) ^b^ 811.7 (181.61) ^c^			102,000 (76,000–123,000) ^b^	124,000 (78,000–156,000) ^b^	856.8 (183.34) ^c^
**Mg**	4000 (2000–7000) ^b^ 389.06 (66.93) ^c^			30,000 (19,000–44,000) ^b^	33,000 (27,000–39,000) ^b^	428.76 (110.42) ^c^

^a^ denotes metal concentrations quantified in [186]. Metals detected by inductively coupled plasma optical emission spectrometry. Data shown as medians (interquartile range). ^b^ denotes metal concentrations quantified in [183]. Metals detected by inductively coupled plasma mass-spectrometry. Data shown as medians (interquartile range). ^c^ denotes metal concentrations quantified in [14]. Metals detected by inductively coupled plasma mass-spectrometry. Data shown as means (standard deviation).

## Data Availability

Not applicable.

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
