# Peer review of "How Does Airway Surface Liquid Composition Vary in Different Pulmonary Diseases, and How Can We Use This Knowledge to Model Microbial Infections?"

_microorganisms, 2024, doi:10.3390/microorganisms12040732_

Round 1
Reviewer 1 Report
Comments and Suggestions for Authors
The review paper by Walsh, Bevan and Harrison addresses the issue of airway surface liquid biochemical composition and how it affects pathological microbes colonization and therefore pathological infections. As the authors point out, the information is relevant to basic biomedical science but also has relevance in the applied biomedical field.
In particular, this information can guide the preparation of specific growth media to better studies in microbiology, in drug discovery, etc…
In biochemical terms, the paper gives a general overview of the biomolecules involved in the airways and how the composition changes during pathological conditions. The role of small molecules (ions, protons (the pH), metals, amino acids, etc..) as well as macromolecular components, like lipids, sugars, DNA, some proteins (albumins, etc..) etc.. have been essentially covered.
However, insights on their interactions have been left uncovered. In our opinion this is a flaw, since not only interaction is of paramount importance in the mucosal physiology but also because, in pure biochemical terms, when preparing growth media, interaction can give rise to a countless series of problems that must be foreseen and avoided.
Another serious flaw, in our opinion, is that the role of microbiota has been left out completely. And the airway microbiota and its activity has fundamental implications both in the composition and the prevention of pathological growths. They can check reviews, such as, for instance:
Li, R., Li, J. & Zhou, X. Lung microbiome: new insights into the pathogenesis of respiratory diseases. Sig Transduct Target Ther 9, 19 (2024). https://doi.org/10.1038/s41392-023-01722-y
in order to get essential information.
We think that without broadening their perspective, the paper could appear as just a limited list of molecules involved in the airway composition. We are convinced that mere knowledge of essential biochemical composition is not sufficient to explain, to model or to prepare relevant growth media to address the role of airway composition in microbial infections.
Author Response
Dear Reviewers,
Thank you all very much for your time reading our manuscript and your constructive comments. Please find our reply to your points below, our responses are highlighted in bold.
Reviewers 1:
The review paper by Walsh, Bevan and Harrison addresses the issue of airway surface liquid biochemical composition and how it affects pathological microbes colonization and therefore pathological infections. As the authors point out, the information is relevant to basic biomedical science but also has relevance in the applied biomedical field.
In particular, this information can guide the preparation of specific growth media to better studies in microbiology, in drug discovery, etc…
In biochemical terms, the paper gives a general overview of the biomolecules involved in the airways and how the composition changes during pathological conditions. The role of small molecules (ions, protons (the pH), metals, amino acids, etc..) as well as macromolecular components, like lipids, sugars, DNA, some proteins (albumins, etc..) etc.. have been essentially covered.
However, insights on their interactions have been left uncovered. In our opinion this is a flaw, since not only interaction is of paramount importance in the mucosal physiology but also because, in pure biochemical terms, when preparing growth media, interaction can give rise to a countless series of problems that must be foreseen and avoided.
- A new section has been added to the manuscript under the subheading ‘Airway surface liquid component interactions’ from line 495 to 541. This new section lays out how various components in an airway surface liquid medium may interact based on previous research into these interactions.
Another serious flaw, in our opinion, is that the role of microbiota has been left out completely. And the airway microbiota and its activity has fundamental implications both in the composition and the prevention of pathological growths. They can check reviews, such as, for instance:
Li, R., Li, J. & Zhou, X. Lung microbiome: new insights into the pathogenesis of respiratory diseases. Sig Transduct Target Ther 9, 19 (2024). https://doi.org/10.1038/s41392-023-01722-y
in order to get essential information.
- A new section has been added under the subheading ‘Lung microbiome’ from lines 543 to 574 and is heavily influenced by the Li et al. (2024) review that you mention. Since the lung microbiome is not a primary focus of this review, this section acts as a short introduction to the lung microbiome. It focuses on how key residents of the lung microbiome change in response to disease, as well as how the lung microbiome in tandem with key airway metabolites has been associated with different clinical outcomes in lung disease. We have directed readers to reviews that comprehensively discuss the importance of the lung microbiome.
We think that without broadening their perspective, the paper could appear as just a limited list of molecules involved in the airway composition. We are convinced that mere knowledge of essential biochemical composition is not sufficient to explain, to model or to prepare relevant growth media to address the role of airway composition in microbial infections.
We hope these revisions are to your satisfaction.
Best wishes,
Dr Dean Walsh, Jennifer Bevan, and Dr Freya Harrison
Reviewer 2 Report
Comments and Suggestions for Authors
Walsh et al. conducted a literature review to discuss and compare the composition - such as lipids, DNA, sugars, metal ions, amino acids, and others - of ASL in different pulmonary pathologies such as Asthma, COPD, Cystic Fibrosis, and Bronchiectasis. The review may appeal to researchers interested in mimicking the respiratory environments to understand and treat pulmonary diseases.
Comment:
The review may benefit from including a discussion and a comparison of microbial flora that colonize the airway surface and whether there are changes in the nature and/or load of this colonization in different diseases.
Author Response
Dear Reviewers,
Thank you all very much for your time reading our manuscript and your constructive comments. Please find our reply to your points below, our responses are highlighted in bold.
Reviewer 2:
Walsh et al. conducted a literature review to discuss and compare the composition - such as lipids, DNA, sugars, metal ions, amino acids, and others - of ASL in different pulmonary pathologies such as Asthma, COPD, Cystic Fibrosis, and Bronchiectasis. The review may appeal to researchers interested in mimicking the respiratory environments to understand and treat pulmonary diseases.
Comment:
The review may benefit from including a discussion and a comparison of microbial flora that colonize the airway surface and whether there are changes in the nature and/or load of this colonization in different diseases.
- To address this we have added a small section titled ‘Lung microbiome’ towards the end of the review (lines 543-574). This serves as an introduction to the importance of the lung microbiome, whilst directed readers to comprehensive reviews on the subject. Within this section, we highlight how the microbiome can change in response to the various lung disease states mentioned earlier in our review.
We hope these revisions are to your satisfaction.
Best wishes,
Dr Dean Walsh, Jennifer Bevan, and Dr Freya Harrison
Reviewer 3 Report
Comments and Suggestions for Authors
Dear editor
The MS entitled “How Does Airway Surface Liquid Composition Vary in Different 2 Pulmonary Diseases, and How Can We Use This Knowledge to Model 3 Microbial Infections?”
It includes the Airway Surface Liquid Composition in relation to microbial infection.
Minor points
-The authors need to add some details about how the bacteria can colonize and evade mucin layer? Role of QS, signaling molecules AHLs and bacterial biofilm formation.
-The authors need to address the effect of pH of fungal colonization or airways.
-What is effect of DNAse producing bacterial such as S. aureus and other Staphylococcus Spp. On DNA layer of airways?
- The effect of some microbial toxins on airways, and how to disseminate.
Comments on the Quality of English Language
Dear editor
The MS entitled “How Does Airway Surface Liquid Composition Vary in Different 2 Pulmonary Diseases, and How Can We Use This Knowledge to Model 3 Microbial Infections?”
It includes the Airway Surface Liquid Composition in relation to microbial infection.
Minor points
-The authors need to add some details about how the bacteria can colonize and evade mucin layer? Role of QS, signaling molecules AHLs and bacterial biofilm formation.
-The authors need to address the effect of pH of fungal colonization or airways.
-What is effect of DNAse producing bacterial such as S. aureus and other Staphylococcus Spp. On DNA layer of airways?
- The effect of some microbial toxins on airways, and how to disseminate.
Author Response
Dear Reviewers,
Thank you all very much for your time reading our manuscript and your constructive comments. Please find our reply to your points below, our responses are highlighted in bold.
Reviewer 3:
Dear editor
The MS entitled “How Does Airway Surface Liquid Composition Vary in Different 2 Pulmonary Diseases, and How Can We Use This Knowledge to Model 3 Microbial Infections?”
It includes the Airway Surface Liquid Composition in relation to microbial infection.
Minor points
-The authors need to add some details about how the bacteria can colonize and evade mucin layer? Role of QS, signalling molecules AHLs and bacterial biofilm formation.
- Lines 140-150 detail how aeruginosa can colonise and evade the mucin layer due to mucin dysfunction in lung disease states such as cystic fibrosis. Lines 150-155 discuss how other lung pathogens navigate the mucin layer.
-The authors need to address the effect of pH of fungal colonization or airways.
- The addition of lines 222-225 highlights the importance of pH manipulation for pathogenesis and colonisation of a variety of fungal species. Unfortunately, from our research we were unable to find examples specific to lung colonisation. With a recent review (ref 118 in text) we cited stating that the effect of pH on fungal colonisation of airways is still largely unexplored.
-What is effect of DNAse producing bacterial such as S. aureus and other Staphylococcus Spp. On DNA layer of airways?
- Lines 196-198 discuss how aureus nucleases degrade DNA in cystic fibrosis sputum to evade neutrophil extracellular traps.
- The effect of some microbial toxins on airways, and how to disseminate.
- Lines 147-150 and 196-198 discuss how different toxins/virulence factors produced by bacterial pathogens are employed in the airways, and their affect on ASL components such as mucin and DNA.
We hope these revisions are to your satisfaction.
Best wishes,
Dr Dean Walsh, Jennifer Bevan, and Dr Freya Harrison
Round 2
Reviewer 1 Report
Comments and Suggestions for Authors
The manuscript has been improved by including two dedicated paragraphs dealing with missing issues in the previous version. Despite the fact that artificially modeling of the airway liquid surface composition is a hard to accomplish feat anyway, the paper has the goal of putting a basis for future studies. In our opinion, the introduction of those two additional issues can be considered a significant improvement for future scientific discussion.